# Genetic and Haplotype Diversity of Manila Clam *Ruditapes philippinarum* in Different Regions of China Based on Three Molecular Markers

**DOI:** 10.3390/ani13182886

**Published:** 2023-09-11

**Authors:** Di Wei, Sichen Zheng, Songlin Wang, Jingkai Yan, Zhihong Liu, Liqing Zhou, Biao Wu, Xiujun Sun

**Affiliations:** 1State Key Laboratory of Mariculture Biobreeding and Sustainable Goods, Yellow Sea Fisheries Research Institute, Chinese Academy of Fishery Sciences, Qingdao 266071, China; weidi@stu.ouc.edu.cn (D.W.); zhengsichen1019@163.com (S.Z.); wslin1126@163.com (S.W.); liuzh@ysfri.ac.cn (Z.L.); zhoulq@ysfri.ac.cn (L.Z.); wubiao@ysfri.ac.cn (B.W.); 2Laboratory for Marine Fisheries Science and Food Production Processes, Laoshan Laboratory, Qingdao 266237, China; 3College of Fisheries, Ocean University of China, Qingdao 260003, China; 4College of Fisheries and Life Science, Shanghai Ocean University, Shanghai 201306, China; 5School of Marine Science and Fisheries, Jiangsu Ocean University, Lianyungang 222005, China; 6Laizhou Marine Development and Fishery Service Center, Yantai 261400, China; xianer1982@163.com

**Keywords:** *Ruditapes philippinarum*, genetic diversity, genetic differentiation, haplotype

## Abstract

**Simple Summary:**

China has the largest production yield of Manila clam *Ruditapes philippinarum* in the world. Most of the clam seeds for aquaculture are mainly derived from artificial breeding in southern China, likely resulting in the loss of genetic variation and inbreeding depression. In this paper, we investigate the genetic diversity and differentiation of *R. philippinarum* populations. The present results not only provide the scientific basis for the phylogenetic relationship of clam resources, but also lay the foundation for natural population conservation and genetic improvement of *R. philippinarum*.

**Abstract:**

China has the largest production yield of Manila clam *Ruditapes philippinarum* in the world. Most of the clam seeds for aquaculture are mainly derived from artificial breeding in southern China, likely resulting in the loss of genetic variation and inbreeding depression. To understand the genetic and haplotype diversity of *R. philippinarum*, 14 clam populations sampled from different regions of China were analyzed by three molecular markers, including *COI*, *16SrRNA* and *ITS*. Based on the results of the *COI* and *ITS* genes, the 14 populations showed a moderate to high level of genetic diversity, with an average haplotype diversity of 0.9242 and nucleotide diversity of 0.05248. AMOVA showed that there was significant genetic differentiation among all populations (mean F_ST_ of the total population was 0.4534). Pairwise F_ST_ analysis showed that genetic differentiation reached significant levels between Laizhou and other populations. Two Laizhou populations showed great divergence from other populations, forming an independent branch in the phylogenetic tree. The shared haplotypes Hap_2 and Hap_4 of *COI* appeared most frequently in most clam populations. In contrast, *16SrRNA* analysis of the clam populations revealed the dominated haplotype Hap_2, accounting for 70% of the total number of individuals. The haplotype diversity of the Laizhou population (Laizhou shell-wide (KK) and Laizhou dock (LZMT)) was relatively higher than other populations, showing multiple unique haplotypes (e.g., Hap_40, Hap_41 and Hap_42). These findings of genetic and haplotype diversity of clam populations provide guiding information for genetic resource conservation and genetic improvement of the commercially important *R. philippinarum*.

## 1. Introduction

Genetic diversity is an essential component of biodiversity, with positive significance for the evolution and conservation of a species [1,2,3,4]. Genetic diversity plays a crucial role in guiding resource management, conservation and assessment of marine organisms [5,6,7]. For decades, molecular markers have become a convenient and effective tool to uncover genetic diversity of aquatic organisms, such as molluscs [3,4,8,9,10]. The application of molecular markers to study genetic diversity in molluscs is important for understanding population genetic structure, inbreeding effects, effective population sizes and the development of conservation strategies in a globally changing environment [11,12,13]. For instance, genetic diversity provides some guiding information on managing natural resources, maintaining sustainable fisheries and selective breeding in the shellfish aquaculture industry [6,14,15]. Due to the small molecular weight and fast evolution, mitochondrial DNA has been widely used in shellfish genetic studies, providing deep insights into molecular phylogenesis, population genetics and conservation genetics [9,16,17,18,19].

The Manila clam *Ruditapes philippinarum*, native to the Pacific and Indian Ocean coast, has been introduced to North America and Europe, becoming a worldwide cultured bivalve in mudflats [20,21]. Due to the delicious taste and *ITS* richness in nutrition, *R. philippinarum* are very much enjoyed by worldwide consumers [10]. In China, *R. philippinarum* becomes one of the most commercially important bivalves from the northern to southern coast, with a high annual production rate (~4 million tons) [22]. For many years, most clam seeds were derived from artificial breeding, produced by a small number of parents in Fujian province, which may result in the loss of genetic diversity and small Ne (effective population sizes) estimates [23,24,25]. Currently, some studies on genetic diversity of *R. philippinarum* have been reported using mitochondrial markers, such as *COI* and 16S rRNA [21,26,27,28,29]. However, most of these studies were sampled from a small number of populations and a single type of molecular markers. The molecular phylogenetics among clam populations remain largely unknown. Therefore, it is essential to investigate the genetic diversity and differentiation of clam populations distributed from the northern to southern coast of China.

Most of the clam seeds for aquaculture are mainly derived from artificial breeding in southern China, likely resulting in the loss of genetic variation. To explore the genetic and haplotype diversity of *R. philippinarum*, 14 clam populations along the coast of China were collected and analyzed by two mitochondrial genes (*COI* and *16SrRNA*) and one nuclear gene (*ITS*). The present results not only provide the scientific basis for the phylogenetic relationship of clam resources, but also lay the foundation for natural population conservation and genetic improvement of *R. philippinarum*.

## 2. Materials and Methods

### 2.1. Sample Collection

Manila clam *R. philippinarum* samples representing 14 populations were taken from the north and south coast of China, including Beihai (BH), Donggang (DG), Dongying (DY), Guangdong (GD), Haiyang (HY), Hongdao (HD), Laizhou shell-wide (KK), Laizhou dock (LZMT), Lianjiang (LJ), Ningbo (NB), Rizhao (RZ), Sanya (SY), Zhangzhou (ZZ) and Zhuanghe (ZH) (Figure 1). Two populations were sampled from Laizhou, including Laizhou shell-wide (KK) and Laizhou dock (LZMT). A total of 20 individuals were used in each population (Table 1).

### 2.2. DNA Extraction, Amplification and Sequencing

The DNA was extracted from the foot muscle of *R. philippinarum* using the standard phenol chloroform method. After DNA extraction, the products were tested by 1% agarose gel electrophoresis for DNA quality. The concentration and purity of DNA was measured using a NanoDrop Lite ultra-micro spectrophotometer (Thermo Scientific, Waltham, MA, USA). The obtained DNA was diluted to 50 ng/μL and stored in an ultra-low-temperature refrigerator at −80 °C.

The universal *COI* primers of LCO1490 and HCO2198 were used for PCR amplification, while *16SrRNA*ar and *16SrRNA*br were used for PCR amplification of *16SrRNA* [30,31]. In addition to *COI* and 16S, the internal transcribed spacer (*ITS*) of nuclear DNA also served as one of the most extensively sequenced molecular markers for investigating genetic diversity among populations. In this study, the primers of *ITS*-F and *ITS*-R were used to amplify the *ITS*2 region [32]. The above primers were synthesized by Sangon Biotech (Shanghai, China).

The polymerase chain reaction (PCR) system (50 µL) consisted of 5 µL of DNA template solution at 50 ng/µL, 25 µL of MIX enzyme (Nanjing Novozymes Biotechnology Co., Ltd., Nanjing, China), 2 µL of each forward and reverse primers and a fixed volume of 50 µL of ultrapure water. The amplification reaction conditions were as follows: pre-denaturation at 94 °C for 5 min; cyclic denaturation at 94 °C for 30 s; annealing and denaturation at 56 °C for 30 s; extension at 72 °C for 1 min for total 35 cycles; and finally, extension at 72 °C for 10 min, ending at 4 °C. Each PCR product was examined on 1% agarose gel to verify the amplified fragment length. Finally, they were sent to Sangon Biotech (Shanghai) Co., Ltd. (Shanghai, China) for Sanger sequencing. In this study, *COI*, *16SrRNA* and *ITS* gene sequences were submitted to the NCBI (https://www.ncbi.nlm.nih.gov/ (accessed on 30 August 2023)) database under accession numbers OR486714—OR486966, OR499223—OR499487 and OR499488—OR499723, respectively.

### 2.3. Data Analysis

The obtained DNA sequences were visualized and checked using SeqMan 7.1.0.44 software [33]. The splicing-corrected DNA sequences were imported into MEGA 11 software for CLUSTAL X homologous sequence comparison to determine the length, genetic distance and base composition of the sequences [34]. Genetic distance analysis was performed in MEGA with 1000 bootstraps replicates. ML analysis was performed in MEGA11 with 1000 bootstraps replicates, and an evolutionary tree was created using the maximum likelihood method [34]. A neighbor-joining (NJ) tree was established based on the genetic distance [35]. Genetic diversity parameters were calculated using DnaSP v5, including the nucleotide polymorphic loci, number of haplotypes, haplotype index and nucleotide diversity index [36]. Molecular analysis of variance (AMOVA) was performed using ARLEQUIN 3.0 software to analyze genetic variation among *R. philippinarum* populations and to calculate the coefficient of genetic differentiation (F_ST_) within and among populations [37]. A haplotype network relationship map was generated using Popart 1.7 based on the TCS method [38].

## 3. Results

### 3.1. Gene Sequence Base Composition of the R. philippinarum Population

The sequences of *COI*, *16SrRNA* and *ITS* genes were used for multiple sequence alignment of the homologous sequences. The primers and partial end sequences were removed. Gene fragments of 559 bp, 431 bp and 349 bp in length were obtained. A total of 253 *COI* sequences, 265 *16SrRNA* sequences and 242 *ITS* sequences derived from 14 populations were compared using MEGA11 software. For *COI*, the average contents of AT and GC were calculated as 66.1% and 33.9%, respectively. The mean contents of *16SrRNA* genes A + T and G + C were estimated to be 67.3% and 32.7%, respectively. In contrast to *COI* and 16S, *ITS* showed the different pattern of AT and GC contents, with AT content (31.6%) significantly lower than GC content (68.4%).

### 3.2. Genetic Diversity Analysis of R. philippinarum Population

The results of genetic diversity based on *COI* genes are shown in Table 1. A total of 70 haplotypes were detected in 253 individuals. The average haplotype diversity of the total population was 0.909, and each population had *ITS* own haplotype. Among them, the largest number of individuals was 14 in the HY and ZH populations, and the smallest number was 6 in the HD population. There were 49 unique haplotypes in the population, accounting for 70% of the total number of haplotypes. There were 21 shared haplotypes, accounting for 30% of the total number of haplotypes. The maximum number of individuals of shared haplotype Hap_2 was 58, accounting for 22.9% of the total number of individuals. The shared haplotype Hap_2 was distributed in 12 populations, except for KK and LZMT. In contrast, some unique haplotypes (Hap_40, Hap_41 and Hap_42) were only distributed in the KK and LZMT populations. A total of 68 nucleotide polymorphic loci were detected in the *COI* gene, including 36 single-mutant loci and 32 parsimony informative loci. The average number of nucleotide differences across all populations was 4.156, and the nucleotide diversity index was 0.00746.

Genetic diversity parameters based on *16SrRNA* gene sequences were summarized in Table 2. The results showed that a total of 24 haplotypes were detected in 265 individual *R. philippinarum*. For 16S, haplotype diversity index was calculated to be 0.488 for all populations. The ZZ population had the most haplotypes (six), and the GD population had the least haplotypes (one). There were 19 population-specific haplotypes, accounting for 79.2% of the total number of haplotypes. There were five shared haplotypes, accounting for 20.8% of the total number of haplotypes. Among them, the maximum number of individuals with shared haplotype Hap_2 was 187, accounting for 70.6% of the total number of individuals. As the dominant haplotype, haplotype Hap_2 was distributed and shared in all populations, except for the DG population. A total of 30 nucleotide polymorphic loci were detected in the *16SrRNA* gene, including 23 single-mutant loci and 7 parsimony informative loci. The average nucleotide difference number of the total population was 0.872, and the nucleotide diversity index was 0.00203. Based on 16S analysis, the two highest numbers of haplotype diversity were found in the LZMT population (0.59) and the ZZ population (0.44), showing relatively higher values than those of the other populations. In contrast, the extremely low level of haplotype diversity (≤0.10) was detected in some populations, including the DG, GD and HD populations.

Genetic diversity parameters based on *ITS* gene sequences showed that a total of 115 haplotypes were detected in 242 individuals (Table 3). The average haplotype diversity index of the total population was 0.9394. The maximum number of haplotypes was 15 for the GD population, and the minimum number was 8 for LZMT. There were 105 population-specific haplotypes, accounting for 91.3% of the total number of haplotypes. There were 10 shared haplotypes, accounting for 8.7% of the total number of haplotypes. Among them, the number of shared haplotype Hap_4 was the largest, with four individuals, accounting for 14.05% of the total number of individuals. The shared haplotype Hap_4 was distributed in all populations, except for the GD population. In contrast, Hap_24-38 were unique haplotypes, which were only distributed in the GD population. A total of 252 nucleotide polymorphic loci were detected in *ITS* genes, including 22 single-mutant loci and 230 parsimony informative loci. The average number of nucleotide differences in the total population was calculated to be 27.493, and the nucleotide diversity index was 0.09749.

The haplotype maps for *COI*, 16S and *ITS* are shown in Figure 2, Figure 3 and Figure 4, respectively. Most of the populations based on the *COI* gene were radially distributed with haplotypes Hap_1, Hap_2 and Hap_4 as the center (Figure 2). The populations based on the *16SrRNA* gene clustered together with haplotype Hap_2 as the center (Figure 3). It can be clearly seen that Hap_2 is the overwhelmingly dominant haplotype, covering the majority of the populations. The haplotype map based on the *ITS* gene showed the unique haplotypes in different populations (Figure 4). The haplotypes were mainly distributed radially with Hap_1, Hap_4, Hap_7 and Hap_8 as the center, showing a high level of haplotype diversity.

### 3.3. Genetic Structure of the R. philippinarum Population

The genetic distances for *COI*, 16S and *ITS* among the 14 populations were shown in Table 4, Table 5 and Table 6, respectively. The genetic distances of *COI*-based calculations ranged from 0.00517 to 0.01377 (Table 4). The largest genetic distance (0.01377) was detected between the KK and RZ population. The smallest genetic distance (0.00517) was found between KK and LZMT. The results of the 16S-based calculations indicated that the genetic distances ranged from 0.000342 to 0.006961 among the populations (Table 5). Similarly, large genetic distances were found between Laizhou and other populations based on the 16S sequences. The largest genetic distance was found to be 0.006961 between DG and LZMT, while the lowest was 0.000342 between the GD and HD populations. The genetic distances based on *ITS* genes ranged from 0.0156 to 0.5255 (Table 6). The largest genetic distance was detected between the GD and SY populations, while the smallest was found between the BH and RZ populations.

The phylogenetic tree of 14 populations constructed using NJ (neighbor-joining method) is shown in Figure 5. The NJ tree based on *COI* and 16S consistently formed into two major branches (Figure 5a,b). KK and LZMT were clustered into one independent branch, while the remaining 12 populations were clustered into one large branch. For *COI*, two northern populations (RZ and HY) and three southern populations (GD, LJ and BH) were grouped into one branch, indicating the close relationship between the northern and southern populations (Figure 5a). The close relationship among the southern and northern populations (e.g., DY, ZH, DG, HY and BH) was also supported by 16S and *ITS*-based NJ tree (Figure 5b,c).

The maximum likelihood (ML) trees for *COI*, 16S and *ITS* are shown in Figure A1, Figure A2 and Figure A3, respectively (Appendix A). The two populations from Laizhou (KK and LZMT) display the independent branch in the phylogenetic tree. The phylogenetic relationship revealed by ML method supports the results of the NJ tree.

The genetic differentiation coefficients among the 14 populations and their significance levels are shown in Table 4, Table 5 and Table 6. The results show that the genetic differentiation index among the 14 populations derived based on *COI* genes ranged from 0.04753 to 0.61486 (Table 4). The smallest genetic differentiation was found between the HD and ZZ populations, while the largest genetic differentiation coefficient was found between BH and LZMT. According to pairwise F_ST_ analysis, genetic differentiation between Laizhou and the other 12 groups was at a highly significant level. The significant differentiation was found between BH and HD, and ZZ and ZH, while non-significant differentiation was detected among the remaining groups. The genetic differentiation based on 16S ranged from −0.0344 to 0.89497 (Table 5). The lowest genetic differentiation was found between HD and NB, while the highest was found between DG and GD. The significant differentiation of 16S was found between the DG and Laizhou populations (KK and LZMT), while others were non-significant. As shown in Table 6, the genetic differentiation based on *ITS* ranged from −0.02152 (between DY and LJ) to 0.96041 (between BH and GD). As indicated by *ITS* analysis, significant differentiation was detected between GD and other populations. The highly significant differentiation was also found between LZMT and BH, HD and RZ, and BH and HY. Furthermore, the significant differentiation appeared among HY and DG and RZ groups, as well as LZ and DG groups.

AMOVA results are summarized for *COI*, 16S and *ITS* in Table 7, Table 8 and Table 9, respectively. For *COI*, the results show that the contribution of genetic variation within the population was 83.73%, while the contribution of genetic variation among populations was 16.27% (Table 7). For 16S, the contribution of genetic variation within the population was 44.06%, while the contribution of genetic variation between populations was 55.94% (Table 8). For *ITS*, the contribution of genetic variation within the *ITS*-based gene population was 36.18%, and the contribution of genetic variation between populations was 63.82% (Table 9). Therefore, *COI*-based genetic differentiation was originated from intra-populations, while 16S- and *ITS*-based genetic differentiation was mainly derived from inter-populations.

## 4. Discussion

### 4.1. Haplotype and Genetic Diversity of R. philippinarum

For animals, a high level of genetic diversity is essential for the long-term survival of populations, having great impacts on their ability to adapt to changing environments [3,39]. DNA barcoding can reveal patterns of genetic diversity by haplotype and nucleotide diversity indexes, representing the two important parameters of genetic diversity in population genetic studies [9]. Haplotype diversity reflects allelic differences among samples, while nucleotide diversity indicates the average number of nucleotide differences among DNA sequences [3]. In the present study, genetic diversity of 14 clam populations revealed by *COI*, *16SrRNA* and *ITS* indicates a pattern of a high level of haplotype and nucleotide diversity in these clam populations. The high level of haplotype and nucleotide diversity (*H_d_* > 0.85 and *P_i_* > 0.05) is comparable with the previous results of other clam populations, indicating large stable populations over time in most clam populations [21,23,25,27,28,40]. High levels of haplotype diversity have been observed in other marine bivalves, such as Caribbean bivalve *Mytilopsis sallei* and mussel *Musculista senhousia* [15,41,42]. A large population size and high-nucleotide-mutation rates may be the main contributors to high genetic diversity [5,43]. In contrast, 16S analysis revealed a low to moderate level of haplotype and nucleotide diversity (*H_d_* < 0.5 and *P_i_* < 0.05) in most studied populations. In particular, the extremely low level of haplotype diversity in some populations suggests that the founder effects may drive the genetic structure of passively dispersed aquatic invertebrates [4,29].

According to the previous study, the low level of haplotype diversity and nucleotide diversity (*H_d_* < 0.5, *P_i_* < 0.005) suggested that a population bottleneck effect or an establishment effect by a single, small population has recently occurred [44]. In the present study, the low level of haplotype and nucleotide diversity was consistently found in all the clam populations, except for LZMT, HY and DY. There are two possible reasons for this phenomenon. One reason could be due to population builder effects in those populations with a low level of diversity [29]. For those populations, the number of nucleotides showing variation is not enough in a short period of time, resulting in the decrease in haplotype and nucleotide diversity. Another possible explanation is that *16SrRNA* was more conserved mitochondrial DNA across species compared to the *COI* and *ITS* genes [29,45,46]. Because the *16SrRNA* gene contains highly conserved regions and highly variable regions, it serves as an effective marker for studying the phylogenetic relationship among species below the family level [21,27,29]. Therefore, these two possible reasons may explain the low level of haplotype and nucleotide diversity in most clam populations.

In the present study, the shared haplotypes Hap_2 and Hap_4 of *COI* appeared most frequently in most clam populations. Moreover, *16SrRNA* analysis of the clam populations revealed the dominate haplotype Hap_2, accounting for 70% of the total number of individuals. The haplotype Hap_2 was distributed in all populations, except for the DG population. One possible reason for this is that the pattern of southern breeding and northern culture in the clam industry may cause high genetic homogeneity [25]. Notably, the haplotype diversity of the LZ population was higher than other sampled populations, showing multiple unique haplotypes in Figure 2 and Figure 3 (e.g., Hap_40, Hap_41 and Hap_42). This may indicate that clams sampled from Laizhou maintain their natural status with high levels of genetic variability, supporting the recent results inferred from microsatellite markers [24,25].

### 4.2. Genetic Differentiation among Populations

Population genetic structure and differentiation are affected by many factors, such as habitat adaptation, natural selection, random genetic drift, and gene flow [8,28,39]. In the present study, significant levels of genetic differentiation were mainly detected between Laizhou and other populations. For instance, genetic differentiation of *COI* was highly significant between Laizhou and other populations, ranging from 0.35 to 0.61 (Table 4). Consistently, genetic differentiation of 16S appeared to be relatively higher than others at highly significant levels, ranging from 0.70 to 0.85 (Table 5). This is also evidenced by the independent branch of the Laizhou populations (KK and LZMT) in the phylogenetic tree (ML and NJ). The great genetic divergence between Laizhou and other populations was also supported by our previous results of microsatellite genotyping [25]. In sharp contrast, non-significant genetic differentiation was more likely to be observed in clam populations, except for Laizhou. These findings support the claim that genetic differentiation among clam populations did not follow the typically spatial distribution characteristics due to geographic isolation [25]. The non-significant genetic differentiation in those non-Laizhou populations suggests the genetic homogeneity of clam populations in the northern and southern coasts. The similar findings were also supported by genetic studies of clam populations using 16S and microsatellite markers [24,25]. A possible explanation for this is that clam seeds’ transplantation among different culture regions may serve as the main contributor for the genetic homogeneity. In recent decades, a large number of clam seeds produced in southern China have been transferred to different culture regions in northern China [47,48]. For *R. philippinarum*, the mode of southern breeding and northern culture may increase the gene flow of clams, resulting in low genetic differentiation between northern and southern populations [24,25,40,47]. In the present study, *COI*, 16S and *ITS* support the conclusion that artificial breeding and culture may serve as one of the major factors influencing the population genetic structure of clams in China.

In the clam aquaculture industry, frequent translocation of artificial clam seeds to natural habitats may result in altering genetic composition in wild populations, even displacing the natural clam populations. The genetic changes affected by clam seed translocation were evidenced by microsatellite marker analysis for western Korean populations [23]. The frequent translocation of clam seeds between different coastal regions may cause a decline in genetic diversity and loss of genetic variation in clams [40]. Moreover, artificial breeding of bivalves in hatcheries may cause small Ne values, resulting in significant inbreeding depression of yield and individual growth rates [48,49]. Therefore, for the sustainable development of clam aquaculture, genetic conservation of wild clam populations is critically important to maintain genetic variability in local populations by conservation programs, such as broodstock management and habitat conservation [7]. Overall, these findings of the genetic and haplotype diversity of *R. philippinarum* will provide some guiding information for the conservation, management and genetic improvement of *R. philippinarum* in China.

## 5. Conclusions

In this study, 14 populations of *R. philippinarum* Manila clam from different regions of China were analyzed by three molecular markers, including *COI*, *16SrRNA* and *ITS*. Based on the results of the *COI* and *ITS* genes, the 14 populations showed a moderate to high level of genetic and haplotype diversity. The shared haplotypes of *COI* and *16SrRNA* were detected in most studied populations, while the LZ populations had some unique haplotypes, showing great divergence from other populations. The present findings support the conclusion that the genetic population structure of clams may be influenced by translocation of artificial clam seeds to different regions for culture in China. The present findings of genetic and haplotype diversity of *R. philippinarum* based on three molecular markers will improve our understanding of the genetic status of clam populations and provide some guiding information for the conservation, management and genetic improvement of *R. philippinarum* in China.

## Figures and Tables

**Figure 1 animals-13-02886-f001:**
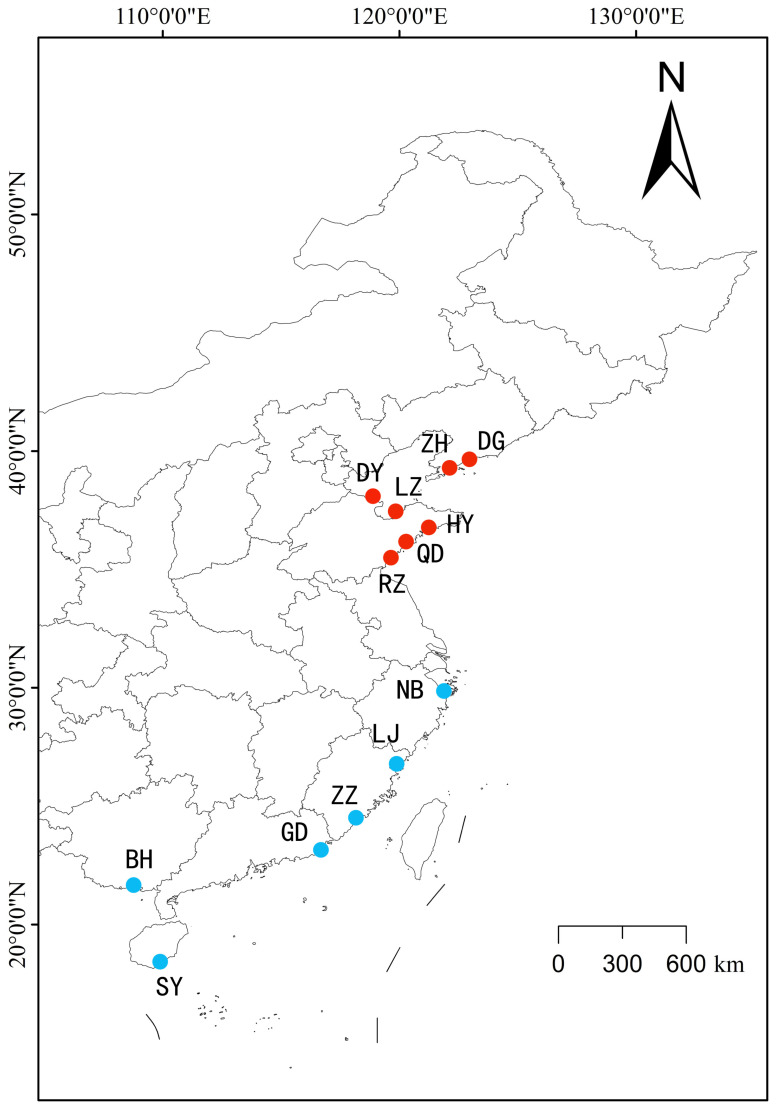
The sampling map for Manila clam *Ruditapes philippinarum* from different locations. Beihai (BH), Donggang (DG), Dongying (DY), Guangdong (GD), Haiyang (HY), Hongdao (HD), Laizhou (LZ), Lianjiang (LJ), Ningbo (NB), Rizhao (RZ), Sanya (SY), Zhangzhou (ZZ) and Zhuanghe (ZH). The red color represents the northern group (DG, ZH, DY, LZ, HY, QD, RZ), and the blue color represents the southern group (NB, LJ, ZZ, GD, BH, SY).

**Figure 2 animals-13-02886-f002:**
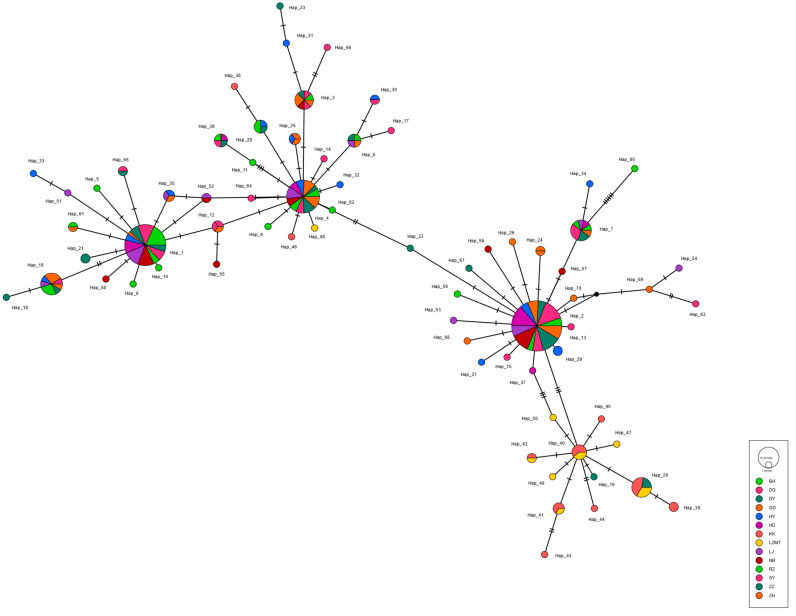
The haplotype map of 14 *R. philippinarum* populations of based on *COI*.

**Figure 3 animals-13-02886-f003:**
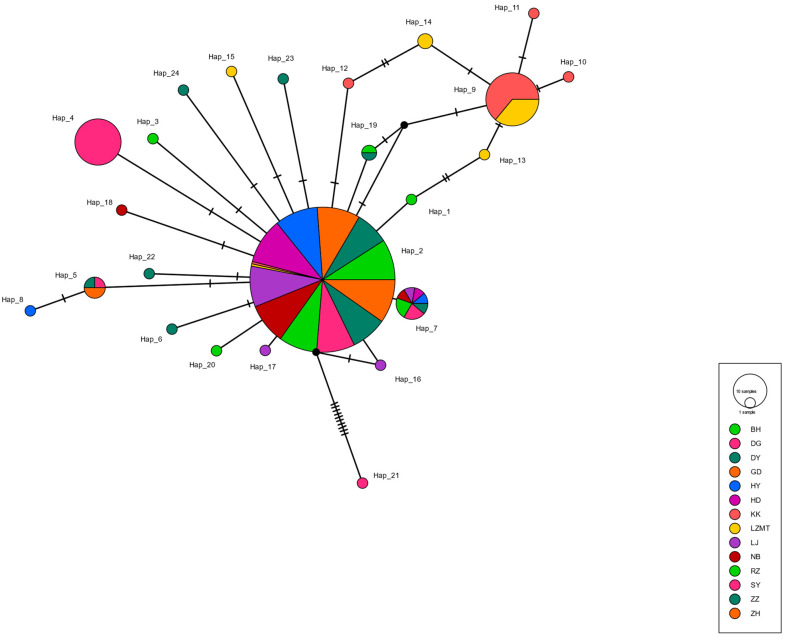
The haplotype map of 14 *R. philippinarum* populations of based on *16SrRNA*.

**Figure 4 animals-13-02886-f004:**
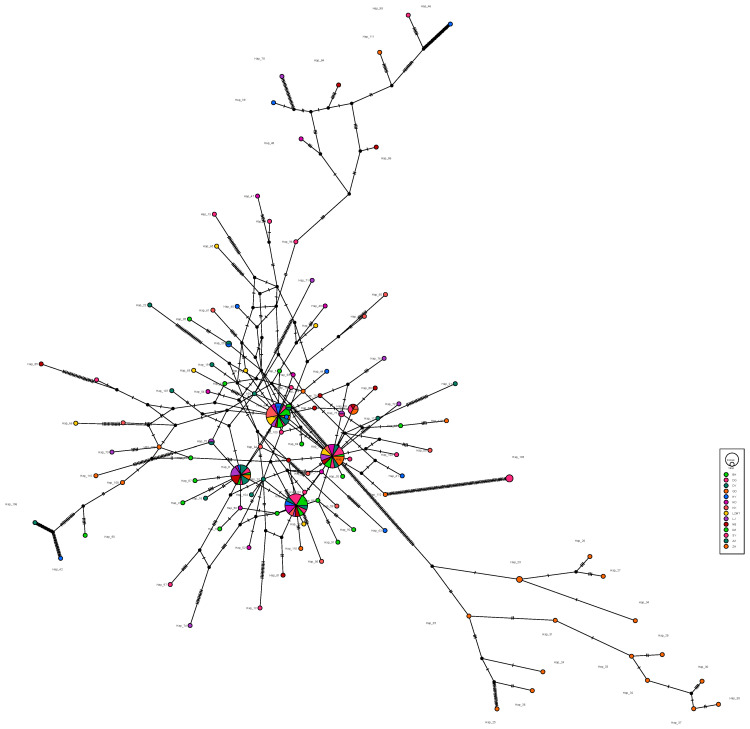
The haplotype map of 14 *R. philippinarum* populations of based on *ITS*.

**Figure 5 animals-13-02886-f005:**
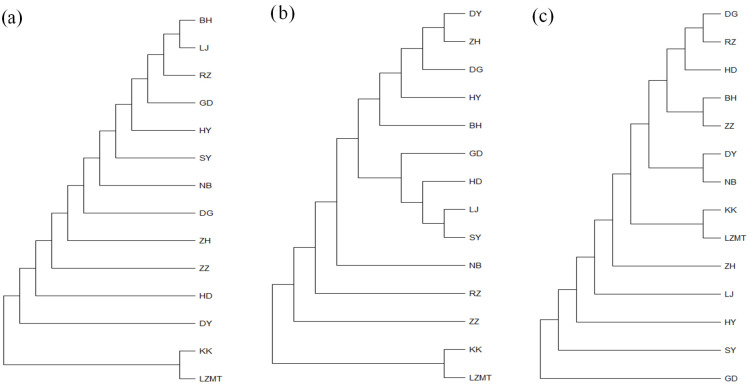
The neighbor-joining (NJ) tree of 14 *R. philippinarum* populations of based on *COI*, *16SrRNA* and *ITS*. (**a**) *COI*-based NJ tree; (**b**) *16SrRNA*-based NJ tree; (**c**) *ITS*-based NJ tree.

**Table 1 animals-13-02886-t001:** Genetic diversity index of 14 populations of *Ruditapes philippinarum* based on *COI* gene sequence and sampling information.

Sample Code	Name	Sample Size	Number of Sequences	Number of Haplotypes	Haplotype Diversity Index	Nucleotide Diversity Index	Average Nucleotide Differences
BH	Beihai	20	20	11	0.86842	0.00514	2.86316
DG	Donggang	20	20	9	0.80000	0.00592	3.30000
DY	Dongying	20	17	11	0.94118	0.00840	4.67647
GD	Guangdong	20	16	8	0.89167	0.00639	3.55833
HY	Haiyang	20	19	14	0.96491	0.00707	3.93567
HD	Hongdao	20	17	6	0.74265	0.00520	2.89706
KK	Laizhou shell-wide	20	17	10	0.91912	0.00610	3.39706
LZMT	Laizhou dock	20	11	8	0.92727	0.00470	2.61818
LJ	Lianjiang	20	20	11	0.90000	0.00610	3.40000
NB	Ningbo	20	20	9	0.83158	0.00542	3.02105
RZ	Rizhao	20	17	11	0.94853	0.00850	4.75529
SY	Sanya	20	20	12	0.93158	0.00719	4.00526
ZZ	Zhangzhou	20	19	9	0.84795	0.00596	3.32164
ZH	Zhuanghe	20	20	14	0.93158	0.00609	3.39474

**Table 2 animals-13-02886-t002:** Genetic diversity index of 14 populations of *Ruditapes philippinarum* based on *16SrRNA* gene sequence.

Population	Number of Sequences	Number of Haplotypes	Haplotype Diversity Index	Nucleotide Diversity Index	Average Nucleotide Differences
BH	19	3	0.20468	0.0049	0.21053
DG	20	2	0.10000	0.0047	0.20000
DY	16	3	0.24167	0.0058	0.25000
GD	18	1	0.00000	0.0000	0.0000
HY	20	3	0.19474	0.0070	0.30000
HD	20	2	0.10000	0.00023	10.000
KK	20	5	0.36842	0.00158	0.67895
LZMT	14	5	0.59341	0.00251	1.07692
LJ	20	4	0.28421	0.0070	0.30000
NB	19	3	0.20468	0.0049	0.21053
RZ	20	4	0.36316	0.00091	0.38947
SY	19	3	0.29240	0.0016	1.35673
ZZ	20	6	0.44737	0.00117	0.50000
ZH	20	2	0.18947	0.00044	0.18947

**Table 3 animals-13-02886-t003:** Genetic diversity index of 14 populations of *Ruditapes philippinarum* based on *ITS* gene sequence.

Population	Number of Sequences	Number of Haplotypes	Haplotype Diversity Index	Nucleotide Diversity Index	Average Nucleotide Differences
BH	18	11	0.89542	0.00994	2.80392
DG	19	9	0.86550	0.01974	5.56725
DY	15	11	0.95238	0.02948	8.31429
GD	16	15	0.99167	0.03183	8.97500
HY	15	12	0.96190	0.07693	21.69524
HD	18	12	0.93464	0.01808	5.09804
KK	20	13	0.91053	0.02714	7.65263
LZMT	13	8	0.88462	0.02741	7.73077
LJ	18	12	0.93464	0.04334	12.22222
NB	18	13	0.94771	0.02603	7.33987
RZ	18	13	0.95425	0.01539	4.33987
SY	19	14	0.96491	0.12986	36.61988
ZZ	16	10	0.93333	0.02813	7.93333
ZH	19	12	0.92398	0.03142	8.85965

**Table 4 animals-13-02886-t004:** Genetic differentiation coefficient (below diagonal) and genetic distance (above diagonal) between populations of clams based on *COI* gene.

	BH	DG	DY	GD	HY	HD	KK	LZMT	LJ	NB	RZ	SY	ZZ	ZH
BH		0.00591	0.00704	0.00601	0.00622	0.00601	0.01331	0.01289	0.00556	0.00548	0.00684	0.00645	0.00622	0.00630
DG	0.06335		0.00722	0.00604	0.00635	0.00540	0.01143	0.01093	0.00593	0.00546	0.00728	0.00645	0.00577	0.00587
DY	0.04202	0.00973		0.00745	0.00771	0.00705	0.01121	0.01070	0.00724	0.00695	0.00853	0.00786	0.00736	0.00749
GD	0.04338	−0.01941	0.00749		0.00650	0.00589	0.01220	0.01174	0.00615	0.00581	0.00727	0.00667	0.00611	0.00613
HY	0.01890	−0.02271	−0.00152	−0.03498		0.00617	0.01238	0.01192	0.00640	0.00608	0.00763	0.00692	0.00642	0.00646
HD	0.13961 *	−0.03027	0.03579	0.01730	0.00487		0.01038	0.00983	0.00583	0.00523	0.00719	0.00614	0.00533	0.00549
KK	0.57961 **	0.47448 **	0.35331 **	0.48834 **	0.46703 **	0.45572 **		0.00517	0.01234	0.01146	0.01377	0.01222	0.01097	0.01123
LZMT	0.61486 **	0.50388 **	0.36994 **	0.51929 **	0.49066 **	0.49308 **	−0.04672		0.01189	0.01098	0.01336	0.01176	0.01047	0.01076
LJ	−0.01182	−0.01438	0.00072	−0.01556	−0.02847	0.03034	0.50555 **	0.53468 **		0.00561	0.00715	0.00658	0.00611	0.00619
NB	0.03695	−0.03965	0.00771	−0.01519	−0.02675	−0.01641	0.49859 **	0.53304 **	−0.02655		0.00699	0.00618	0.00558	0.00569
RZ	0.00562	0.01118	0.00947	−0.02491	−0.02017	0.04760	0.46995 **	0.48764 **	−0.01939	0.00620		0.00780	0.00733	0.00746
SY	0.04340	−0.01745	0.00955	−0.01932	−0.03060	−0.01028	0.45446 **	0.47639 **	−0.01096	−0.01978	−0.00529		0.00641	0.00648
ZZ	0.10741 *	−0.03039	0.02568	−0.01007	−0.01467	−0.04753	0.45050 **	0.48127 **	0.01225	−0.01944	0.01474	−0.02607		0.00581
ZH	0.10856 *	−0.02444	0.03416	−0.01850	−0.01869	−0.03001	0.45718 **	0.48734 **	0.01454	−0.01275	0.02362	−0.02493	−0.03835	

Note: “*” indicates that F_ST_ reached a significant level (*p* < 0.05); “**” indicates that F_ST_ reached a highly significant level (*p* < 0.01).

**Table 5 animals-13-02886-t005:** Genetic differentiation coefficient (below diagonal) and genetic distance (above diagonal) between populations of clams based on *16SrRNA* gene.

	BH	DG	DY	GD	HY	HD	KK	LZMT	LJ	NB	RZ	SY	ZZ	ZH
BH		0.002687	0.000656	0.000656	0.000714	0.000598	0.004891	0.004989	0.000830	0.000611	0.000830	0.002079	0.000946	0.000592
DG	0.81439 **		0.002596	0.002449	0.002657	0.002552	0.006845	0.006961	0.002784	0.002564	0.002784	0.004033	0.002900	0.002529
DY	0.00088	0.80105 **		0.000419	0.000624	0.000522	0.004814	0.004930	0.000754	0.000534	0.000754	0.002003	0.000870	0.000493
GD	−0.00293	0.89497 **	0.00758		0.000470	0.000342	0.004653	0.004769	0.000580	0.000373	0.000593	0.001821	0.000709	0.000361
HY	−0.00047	0.78166 **	−0.02422	−0.00544		0.000563	0.004873	0.004989	0.000795	0.000580	0.000789	0.002030	0.000917	0.000557
HD	0.00097	0.85714 **	0.00577	−0.00544	−0.02564		0.004757	0.004873	0.000667	0.000464	0.000673	0.001896	0.000801	0.000464
KK	0.78114 **	0.85103 **	0.76653 **	0.81782 **	0.76692 **	0.80526 **		0.001972	0.004988	0.004769	0.004884	0.006238	0.005000	0.004757
LZMT	0.71819 **	0.80999 **	0.69803 **	0.75908 **	0.70484 **	0.74768 **	−0.02881		0.005104	0.004885	0.005005	0.006354	0.005121	0.004873
LJ	−0.00047	0.78261 **	−0.00116	−0.00544	−0.01695	−0.02564	0.76692 **	0.70484 **		0.000696	0.000905	0.002128	0.001032	0.000696
NB	0.00000	0.81439 **	0.00088	−0.00293	−0.02144	−0.03440	0.78114 **	0.71922 **	−0.02144		0.000684	0.001931	0.000812	0.000476
RZ	0.01633	0.75439 **	0.01314	0.01980	−0.01393	−0.01974	0.74622 **	0.68159 **	−0.01393	−0.01854		0.002128	0.001009	0.000696
SY	0.00741	0.54331**	0.00026	0.00543	−0.00401	−0.00421	0.61588 **	0.54166 **	−0.00728	−0.00672	−0.01014		0.002268	0.001945
ZZ	−0.00109	0.72000 **	−0.00414	−0.00544	−0.01266	−0.01695	0.72646 **	0.65992 **	−0.01266	−0.01599	−0.02238	−0.00385		0.000812
ZH	0.02581	0.82134 **	−0.03271	0.04509	−0.01974	0.03509	0.78819 **	0.72861 **	0.02105	0.02581	0.03509	0.01648	0.01504	

Note: “**” indicates that F_ST_ reached a highly significant level (*p* < 0.01).

**Table 6 animals-13-02886-t006:** Genetic differentiation coefficient (below diagonal) and genetic distance (above diagonal) between populations of clams based on *ITS* gene.

	BH	DG	DY	GD	HY	HD	LZ	LZMT	LJ	NB	RZ	SY	ZZ	ZH
BH		0.0177	0.0224	0.5036	0.0477	0.0188	0.0228	0.0232	0.0305	0.0210	0.0156	0.0810	0.0223	0.0247
DG	0.00077		0.0258	0.5062	0.0511	0.0222	0.0266	0.0275	0.0342	0.0249	0.0192	0.0849	0.0265	0.0286
DY	0.00389	−0.00231		0.5078	0.0545	0.0265	0.0309	0.0315	0.0377	0.0289	0.0239	0.0892	0.0304	0.0328
GD	0.96041 **	0.95073 **	0.94023 **		0.5134	0.5054	0.5082	0.5088	0.5097	0.5063	0.5068	0.5255	0.5064	0.5089
HY	0.05375 **	0.03393 *	−0.00240	0.89671 **		0.0513	0.0550	0.0560	0.0620	0.0534	0.0487	0.1107	0.0556	0.0566
HD	0.01454	−0.01671	−0.00551	0.95207 **	0.01638		0.0274	0.0279	0.0347	0.0256	0.0202	0.0856	0.0271	0.0293
LZ	0.03103 *	0.01026	0.00238	0.94325 **	0.01662	0.00992		0.0316	0.0388	0.0296	0.0245	0.0890	0.0315	0.0331
LZMT	0.07220 **	0.04760 *	0.01877	0.94196 **	0.01538	0.05842 **	0.02937 *		0.0394	0.0307	0.0250	0.0903	0.0316	0.0335
LJ	0.00400	0.00707	−0.02152	0.92652 **	−0.00184	−0.00031	0.00625	0.01056		0.0371	0.0319	0.0960	0.0384	0.0405
NB	0.00164	−0.00466	−0.01783	0.94399 **	0.00440	−0.01355	−0.00632	0.03024	−0.01248		0.0229	0.0875	0.0297	0.0316
RZ	0.01507	0.00191	0.01419	0.95490 **	0.03070 *	−0.00127	0.01878	0.05320 **	0.00554	0.00676		0.0828	0.0242	0.0259
SY	0.08917	0.08211	0.06136	0.84080 **	0.03720	0.08015	0.07718	0.06848	0.05694	0.06890	0.08312		0.0893	0.0906
ZZ	−0.00058	−0.00320	−0.02070	0.94163 **	0.01622	−0.00576	0.01419	0.02137	−0.02041	−0.00815	0.00194	0.06673		0.0333
ZH	0.01815	0.01137	−0.00262	0.93873 **	0.00082	0.00220	0.00581	0.01076	−0.00737	−0.00905	−0.00905	0.06779	0.00196	

Note: “*” indicates that F_ST_ reached a significant level (*p* < 0.05); “**” indicates that F_ST_ reached a highly significant level (*p* < 0.01).

**Table 7 animals-13-02886-t007:** Analysis of molecular variance of (AMOVA) all the 14 populations of *Ruditapes philippinarum* based on *COI* gene sequence.

Source of Variation	Degree of Freedom	Sum of Squares	Variance Component	Percentage of Variation (%)
Among populations	13	103.073	0.34184 Va	16.27
Within populations	239	420.591	1.75980 Vb	83.73
Total	252	523.664	2.10163	100
Total genetic differentiation index (F_ST_): 0.16265				

**Table 8 animals-13-02886-t008:** Analysis of molecular variance of (AMOVA) all the 14 populations of *Ruditapes philippinarum* based on *16SrRNA* gene sequence.

Source of Variation	Degree of Freedom	Sum of Squares	Variance Component	Percentage of Variation (%)
Among populations	13	64.935	0.25350 Va	55.94
Within populations	251	50.125	0.19970 Vb	44.06
Total	264	115.060	0.45321	100
Total genetic differentiation index (F_ST_): 0.55936				

**Table 9 animals-13-02886-t009:** Analysis of molecular variance of (AMOVA) all the 14 populations of *Ruditapes philippinarum* based on *ITS* gene sequence.

Source of Variation	Degree of Freedom	Sum of Squares	Variance Component	Percentage of Variation (%)
Among populations	13	2127.224	9.17407 Va	63.82
Within populations	228	1185.718	5.20052 Vb	36.18
Total	241	3312.942	14.37459	100
Total genetic differentiation index (F_ST_): 0.63821				

## Data Availability

Not applicable.

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
