# Peer review of "Genetic and Haplotype Diversity of Manila Clam Ruditapes philippinarum in Different Regions of China Based on Three Molecular Markers"

_animals, 2023, doi:10.3390/ani13182886_

Round 1

Reviewer 1 Report

The authors successfully provided valuable information on genetic diversity of the commercially important Manila clam Ruditapes philippinarum in the coasts of China, through a genetic variation analysis at two mitochondrial genes and one nuclear gene adding new insights into the differentiation of the Manila clam population in Laizhou.

In all, the impression is positive, and the methods were presented clearly. However, results are not always written in a simple and easily digestible way, and the discussion tends to repeat the results losing the key message that the authors want to give. Therefore, although I enjoyed reading the manuscript there are some missing information that must be considered before publication.

1.       The title should include the region in which the species are studied

2.       Lines 43-45: Please rephase here without repeat “important”

3.     Lines 71-75: Please rephase this crucial part of the paper and explain the purpose of the study clearly (what are your hypothesis? what is the problem and what did you do to solve this problem?)

4.    Line 80: here you indicated Laizhou shell-wide (KK) and Laizhou dock (LZMT), but in Table 1 KK becomes “Kekuan” and LZMT corresponds to “Laizhoumatou”. Please use the same name locality.

5.       Figure 1: please add legend with name localities and explain different colors.

6.   Line 93-98: here the sentences are incomplete, I suggest something as: “The universal mitochondrial primers LCO1490 and HCO2198 [32] were used for the PCR amplification of the COI gene, while for the 16SrRNA gene we used the primers 16SrRNAar and 16SrRNAbr [33]. The nuclear gene ITS was amplified using the primers ITS-F and ITS-R (ADD REFERENCE)

7.  Line 110: please add the reference for MEGA11 software.

8.  Line 112: why only Neighbor-joining tree? If you want to deepen phylogenetic relationships among populations, you should add also a Bayesian approach and a Maximum Likelihood tree. All these methods detect structure and if all methods produce similar results, you can be quite confident about your results. Concerning the NJ, please add a bootstrap value for each node, and the outgroup.

9. Results: try not to repeat what you have already written in the Materials and methods (for instance, the sentence in line 132-134 "The aligned sequences of each gene were imported into DNASP software to calculate the genetic diversity parameters of each population. Genetic diversity parameters were calculated based on COI gene sequences" can be removed or rephrased). Also, for some aspect results are written in a difficult and repetitive way, please check and try to merge information clearly and easily.

 10.  Discussion: also here, try to not repeat what you have already written in the Results but focus on the important key message of the work.

Author Response

Dear Editor, Thank you very much for the useful suggestions and constructive comments on the manuscript. We have carefully revised the manuscript using highlighted text according to all the suggestions.Please see the attachment.

Reviewer 2 Report

General concept comments:

Judging in general by all the information provided by the authors, the purpose of the article is to additionally analyze the Genetic and haplotype diversity of Manila clam Ruditapes philippinarum (Population Genetics of Manila Clam (Ruditapes philippinarum) in China Inferred from Microsatellite Markers) already studied by them earlier. Additional genetic markers (mtDNA and a polymorphic fragment of the nuclear genome - three molecular markers) are used in addition to the previously used microsatellites
The authors mention this previously published article in almost all sections of the manuscript, but do not specifically discuss what additional information they obtained using additional genetic markers. In other words, the authors have done a thorough analysis, but the nature of the presentation (discussion of the results) adds little information to previous studies.

Some specific comments

Too many tables...

Map (Figure 1 line 85):

- There are errors in the map (not all collection points are marked)

- No explanation for color coding.

- It would be good, if possible, to note artificially cultivated and wild (if any) populations.

Markers:

- There is no information about placing sequences in any database.

- Nowhere in the text is there an explanation of what ITS is and which spacer was used in this analysis.

Networks:

- CO1 and 16S are both mtDNA fragments. For them, you can build a common network and thus increase the resolution.

- Poor quality of the drawing (figure 4). Very small text.

- The color scheme in the designations of the samples in the networks was unsuccessfully chosen - visually, some colors match.

- Apparently there are errors in the designation of haplotypes in networks. Hapl 40, 41, 42 are mentioned even in the abstract as unique for one of the populations, but are actually present in 2 (lines 143 and 37).

Phylogenetic schemes:

- Only one algorithm was used to construct phylogenetic schemes (NJ).

- Which Nucleotide substitution model was used?

- Statistical tests of tree obtained are absent.

- Why this visualization option was used (topology only).

- To facilitate perception, it would be nice to designate the northern and southern populations on phylogenetic schemes (in accordance with the map - Figure 1).

Author Response

Dear Editor,

Thank you very much for the useful suggestions and constructive comments on the manuscript. We have carefully revised the manuscript using highlighted text according to all the suggestions. Please see the attachment.

Reviewer 3 Report

The present manuscript deals with an assay of genetic diversity in Manila clam populations. It's scientifically sound, I have no comments on that. My major concern is on Discussion section. In my opinion, it's a bit short and you have to analyze more your interesting findings. For example, you should focus (by discussing other papers from literature) on the different levels of genetic diversity found by 16S gene compared to COI & ITS genes. What are your speculations/suggestions on this difference??

You can also discuss more the fact that your populations are 'healthy' in terms of genetic variation (no significant inbreeding found). How do you explain that??? You expected (as mentioned in introduction) that your populations would suffer from inbreeding... 

Minor editing of English language required

Author Response

(The authors gave the same response as above.)

Round 2

Reviewer 1 Report

I only have a suggestion, if possible, please move the new 3 figures in the Supplementary material.

Author Response

Dear Editor,

Thank you very much for the useful suggestions and constructive comments on the manuscript. We have carefully revised the manuscript using highlighted text according to all the suggestions. A point-by-point description of revisions made in response to the suggestions is as follows:
